

# Combining multi-dimensional data to identify key genes and pathways in gastric cancer

Wu Ren[1,2], Wei Li[1], Daguang Wang[1], Shuofeng Hu[2], Jian Suo[1] and Xiaomin Ying[2]

[1] Department of Gastrointestinal Surgery, The First Hospital of Jilin University, Changchun, China
[2] Beijing Institute of Basic Medical Sciences, Beijing, China

## ABSTRACT

Gastric cancer is an aggressive cancer that is often diagnosed late. Early detection and treatment require a better understanding of the molecular pathology of the disease. The present study combined data on gene expression and regulatory levels (microRNA, methylation, copy number) with the aim of identifying key genes and pathways for gastric cancer. Data used in this study was retrieved from The Cancer Genomic Atlas. Differential analyses between gastric cancer and normal tissues were carried out using Limma. Copy number alterations were identified for tumor samples. Bimodal filtering of differentially expressed genes (DEGs) based on regulatory changes was performed to identify candidate genes. Protein–protein interaction networks for candidate genes were generated by Cytoscape software. Gene ontology and pathway analyses were performed, and disease-associated network was constructed using the Agilent literature search plugin on Cytoscape. In total, we identified 3602 DEGs, 251 differentially expressed microRNAs, 604 differential methylation-sites, and 52 copy number altered regions. Three groups of candidate genes controlled by different regulatory mechanisms were screened out. Interaction networks for candidate genes were constructed consisting of 415, 228, and 233 genes, respectively, all of which were enriched in cell cycle, P53 signaling, DNA replication, viral carcinogenesis, HTLV-1 infection, and progesterone mediated oocyte maturation pathways. Nine hub genes (SRC, KAT2B, NR3C1, CDK6, MCM2, PRKDC, BLM, CCNE1, PARK2) were identified that were presumed to be key regulators of the networks; seven of these were shown to be implicated in gastric cancer through disease-associated network construction. The genes and pathways identified in our study may play pivotal roles in gastric carcinogenesis and have clinical significance.

# INTRODUCTION

Gastric cancer is the fifth most common cancer and third leading cause of cancer-related death in the world, accounting for an estimated 951,600 new cases and 723,100 deaths in 2012 (*Torre et al., 2015*). Symptoms usually appear late in the disease course and most patients are diagnosed at an advanced stage, which contributes to poor prognosis (*McLean & El-Omar, 2014*). Despite an overall decline in incidence and mortality rate due to advances in our understanding of the disease, gastric cancer remains a significant

Corresponding authors
Jian Suo, suojian0431@hotmail.com
Xiaomin Ying, yingxm@bmi.ac.cn

health care burden worldwide (*Van Cutsem et al., 2016*). Early detection and treatment are essential for improving gastric cancer outcome; this requires a better understanding of the molecular pathology of the disease as well as identification of appropriate biomarkers and drug targets.

Cancer is a complex disease that involves dysregulation at multiple levels. Changes in gene expression resulting from alterations in microRNA (miRNA) expression, DNA methylation level, and DNA copy number have been implicated in gastric carcinogenesis. MiRNAs regulate target mRNA translation and degradation by binding to the 3′ untranslated region (*Tchernitsa et al., 2010*). Several miRNAs have been linked to gastric cancer progression and prognosis (*Ueda et al., 2010*; *Xu et al., 2012*). DNA methylation is another regulatory mechanism implicated in tumor development; loss or gain of methylation marks at CpG sites can activate oncogenes or inactivate tumor suppressor genes. Silencing of PTEN and FOXD3 as a result of hypermethylation has been identified in gastric cancer (*Kang, Lee & Kim, 2002*; *Cheng et al., 2013*). Genomic instability due to copy number alteration (CNA) is also associated with altered gene expression in carcinogenesis (*Liang, Fang & Xu, 2016*). For example, amplification of 8q24, 10q26, 11p13, and 20q13 and deletion of 5p15 and 9p21—which encompass the MYC, FGFR2, and IRX1 loci—have been reported in gastric cancer (*Guo et al., 2010*; *Deng et al., 2012*; *De Souza et al., 2013*).

Differentially expressed genes (DEGs) can be identified by expression profiling, but most of them may have non-essential roles in carcinogenesis (*Fan et al., 2012*). Research on critical biological process and molecule identification should take into consideration the interaction of multiple factors and regulatory mechanisms. However, this presents a challenge in terms of combining data from different experiments to generate biological meaningful and experimentally testable models. A modified network-based method that analyzes individual genes and their interactions as well as upstream regulatory mechanisms is an effective approach (*Ping et al., 2015*; *Rajamani & Bhasin, 2016*). While interaction networks can reveal links between genes and pathways; incorporating regulatory change can provide additional insight for their dysregulation (*Chuang et al., 2007*).

The Cancer Genome Atlas (TCGA) dataset contains cancer-related omics profiles that are useful for systems biology-based investigations of gastric cancer. In the present study, we used data on gene expression and regulatory levels (miRNA expression, DNA methylation, DNA copy number) of stomach adenocarcinoma (STAD) obtained from TCGA to carry out a multi-dimensional analysis as well as regulatory interaction filtering and network analysis to identify key genes and pathways in gastric cancer.

## METHODS

### Data retrieval

Multi-dimensional data (level 3) for gastric cancer were derived from TCGA STAD cohort at Broad GDAC Firehose data run (version: 2016_01_28, http://gdac.broadinstitute.org/). A total of 272 patients for whom gene expression, miRNA expression, and copy number profiles were available were included in the analysis (Table S1). Gene expression profiles of 29 paired tumor and normal tissue samples were measured as a reads per kilobase

**Table 1 Multi-dimensional data for gastric cancer used in this study.**

| Type | Platform/category | Number[a] | Paired sample |
|------|-------------------|-----------|---------------|
| Gene | Illumina Hiseq 2000 + GA | 272 | 29 |
| MiRNA | Illumina Hiseq 2000 + GA | 272 | 34 |
| Methylation | Illumina Infinium HumanMethylation27 | 43 | 22 |
| Copy number | Affymetrix SNP 6.0 | 272 | |

**Notes.**
[a] Number of patients.

per million mapped reads value, and miRNA expression profiles of 34 paired tumor and normal tissue samples were measured as a reads per million value. DNA copy number profiles for 272 patients were obtained from the Affymetrix SNP6.0 platform and processed using the circular binary segmentation method. DNA methylation profiles for 22 paired tumor and normal tissue samples were also included in the study, which were obtained from the Infinium HumanMethylation27 platform (Illumina, San Diego, CA, USA) and shown as a beta value. The Infinium HumanMethylation27 array covered 27,578 CpG sites in 14,495 human genes. The methylation value for a specific gene site was measured by calculating the mean value of all related probes. Detailed information regarding the data is shown in Table 1.

## Identification of alterations at multiple levels

The Limma package in R software was used to detect differentially expressed genes or miRNAs or differential methylation between paired gastric adenocarcinoma and normal tissue samples. Those that met the criteria of fold change $\geq 2$ and Benjamini–Hochberg correlated $P$ value $<0.01$ were considered significant. The unsupervised hierarchical cluster analysis was performed using R gplots package. For somatic copy number data, we used genomic regions with statistically significant focal copy number changes 2.0 (GISTIC2.0) module of the GenePattern public server to identify chromosome regions and genes that were amplified or deleted (*Mermel et al., 2011*). GISTIC2.0 uses ratios of segmented tumor copy number data relative to normal samples as input, and segmented level 3 data were aligned to Hg19 for analysis runs. A cutoff $q$ value of 0.01 was applied to significant loci and genes. Five types of copy number calls (homozygous deletion, heterozygous deletion, diploid, gain, and amplification) were determined for each gene in all cancer samples; only amplification and homozygous deletions were regarded as significant changes in a sample.

## MiRNA-target gene interaction

MiRNA-gene interactions were predicted using Starbase 2.0, which included the TargetScan, PicTar, RNA22, PITA, and miRanda algorithms (*Yang et al., 2011*). Among the miRNA-target gene pairs, only those predicted by at least three algorithms were selected. To identify functional pairs, we also calculated Pearson's correlation coefficient between miRNA and target gene expression for all 272 patients using the cor function in R software (*R Core Team, 2015*).

## Bimodal filtering of differentially expressed genes

To clarify the cross-talks between gene expression and regulatory changes, we filtered out their regulatory interactions. For miRNAs, genes identified as differentially expressed were compared to miRNA targets, with up- and down-regulated miRNAs corresponding to down- and up-regulated genes, respectively. A similar analytical approach was used to assess regulatory interactions between differentially expressed and methylated genes as well as those with CNAs. These DEGs whose expression may be affected by regulatory changes were identified as candidate genes. The correlation between gene expression and copy number was also calculated with the cor function in R software.

## Functional enrichment analysis

Gene function annotation was performed using the Database for Annotation, Visualization, and Integrated Discovery v.6.8 (DAVID v.6.8) to complement Kyoto Encyclopedia of Genes and Genomes (KEGG) pathway and gene ontology (GO) analyses. Significant terms were filtered out with a corrected $P$ value <0.05 (Benjamini–Hochberg method). The GO analysis was limited to biological process terms; organization and visualization were carried out using the Enrichment Map plugin on the Cytoscape platform. GO terms were connected based on their overlap of shared genes and grouped by functional similarity.

## Network construction and analysis

Candidate genes were used to generate interaction networks under regulatory mechanisms. Information on protein-protein interactions (PPIs) was derived from Search Tool for the Retrieval of Interacting Genes/Proteins v.10 (STRING v.10). Only experimentally validated interactions with a score ≥0.4 were used. Networks were generated on Cytoscape software as follows: (i) interaction networks were constructed for DEGs based on protein interconnection information; and (ii) candidate (seed) genes were extracted along with their first interacting neighbors from the DEG network to reconstruct a new subnetwork, respectively. The Network Analyzer plugin on Cytoscape was used for topological analysis. The parameters of degree, betweenness, and closeness—the most important topological parameters were analyzed. The top 10 ranked nodes for each parameter were kept. And genes that ranked top 10 under at least two parameters were considered as hubs.

A disease-associated gene interaction network for gastric cancer was constructed using the Agilent literature search plugin on Cytoscape, which provided interactions reported in the literature. Gene names were used as input, and the disease name was used as the context in the literature search to generate the network.

## RESULTS

### Transcriptome alterations and functional enrichment analysis

We identified 3,602 genes (861 down-regulated and 2,741 up-regulated) that were differently expressed between paired gastric cancer and normal tissue samples (Table S2). Unsupervised clustering divided samples into tumor and normal subgroups according to expression of these genes (Fig. 1A). The GO analysis revealed that these genes were enriched in 62 significant terms which were mainly grouped in clusters of cell cycle, chromatin organization, catabolic process and DNA biogenesis. Specifically, these genes

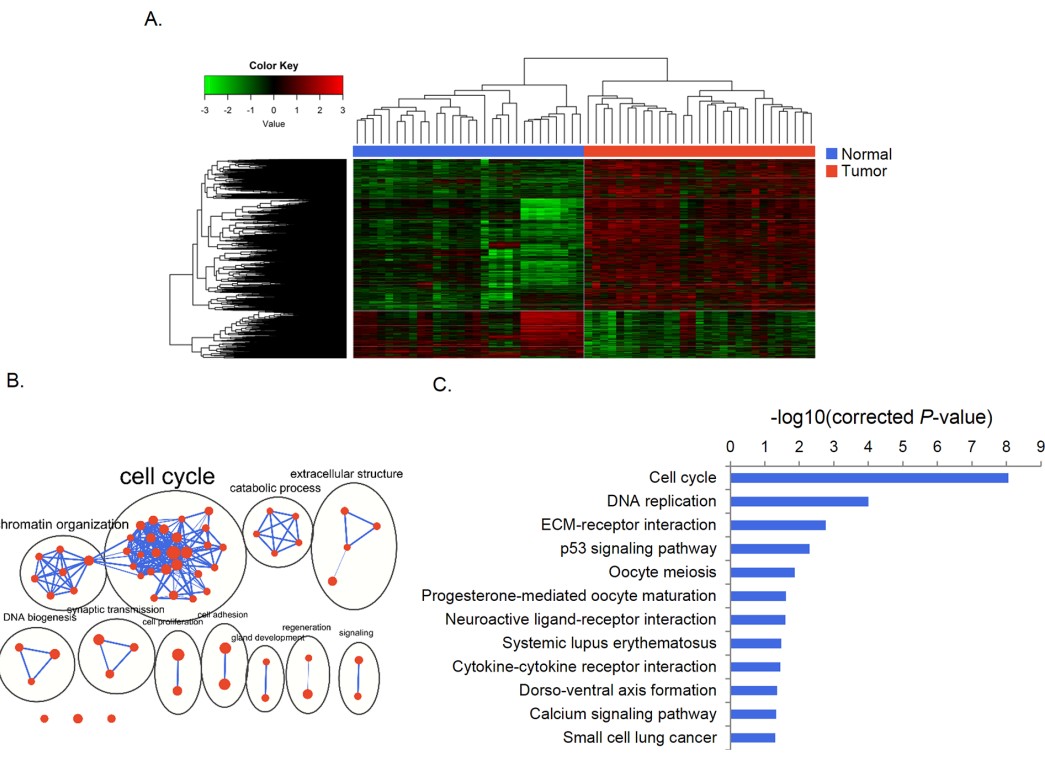

**Figure 1** **Differential gene expression and functional enrichment analysis.** (A) Heat map generated by unsupervised clustering, which divided samples into tumor and normal groups according to DEGs. (B) Enrichment map of GO terms. Nodes represent GO terms, which were clustered and annotated by their similarity. (C) KEGG pathway analysis of DEGs.

were linked to cell cycle, DNA replication, ECM-receptor interaction, p53 signaling pathways (Figs. 1B, 1C).

## Identification of alterations at regulatory levels

A total of 251 miRNAs were differentially expressed (20 down-regulated and 231 up-regulated) between 34 paired tumor and normal tissue samples. Additionally, 604 genes were found to be differentially methylated (206 and 398 that were hyper- and hypo-methylated, respectively), and 52 chromosomal regions were altered (21 amplifications and 31 deletions) in 272 gastric cancer patients, as determined by the GISTIC2.0 algorithm (Fig. 2). A total of 331 and 1,806 target genes were located within these amplified and deleted regions, respectively. These alterations were considered as multi-dimensional signatures for gastric cancer (Table S2).

## Regulatory interactions and candidate genes under regulatory control

We then examined the occurrence of regulatory interactions between gene expression and regulatory changes. The cross-talk between differentially expressed genes and miRNAs was detected on the basis of miRNA-target genes. A total of 82 altered miRNAs were found to regulate 514 DEGs (Fig. 3A); of these, 11 down-regulated miRNAs targeting 100 up-regulated genes and 65 up-regulated miRNAs targeting 112 down-regulated genes were

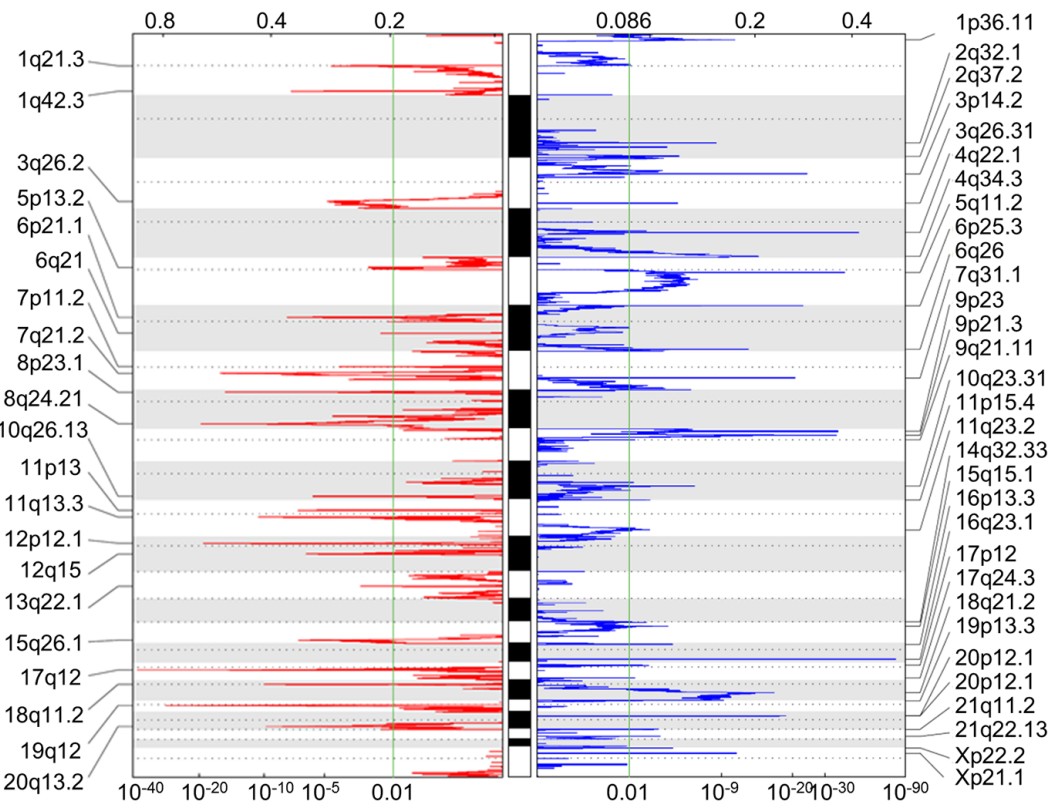

**Figure 2** **Somatic CNA data of focal amplifications and deletions.** GISTIC2.0 identified 21 amplified and 31 deleted focal regions in segmented single nucleotide polymorphism array data of 272 STAD patients. Normalized amplifications and deletions are shown in red and blue, respectively.

screened. The same analytical process was applied to methylation and CNA; there were 74 genes that were overexpressed and hypomethylated and 13 that were underexpressed and hypermethylated; and 47 and 66 genes that were over- and under-expressed, respectively, and had CNAs. These DEGs were regarded as genes under different regulatory controls and taken as candidate genes (Fig. 3B, Table S3). When these candidate genes were categorized into three groups according to regulatory mechanisms, some overlap between the groups was noted. A total of nine and 16 up- and down-regulated genes, respectively, were common to at least two groups. Only one gene, MYEF2, belonged to all three groups, and was found to be down-regulated and hypermethylated with reduced copy number.

The correlation between gene expression and regulatory controls was also determined for candidate genes in the miRNA and CNA groups. Inversely correlated miRNA-gene pairs were kept, leaving 467 functional pairs including 67 miRNAs and 151 genes with a criteria of $r < -0.1$ and $P < 0.05$. MiR-92a-3p, miR-19a-3p, and miR-19b-3p—all of which belong to the miR-17-92 oncomiR cluster—had the highest number of significant targets (Fig. 3C). These miRNAs are known to promote cell proliferation, metastasis, and drug resistance in gastric cancer (Wang et al., 2013; Wu et al., 2013; Wu et al., 2014). To evaluate the contribution of CNAs to transcriptomic changes, we performed a correlation analysis between copy number and candidate gene expression. The remaining 11 genes were with

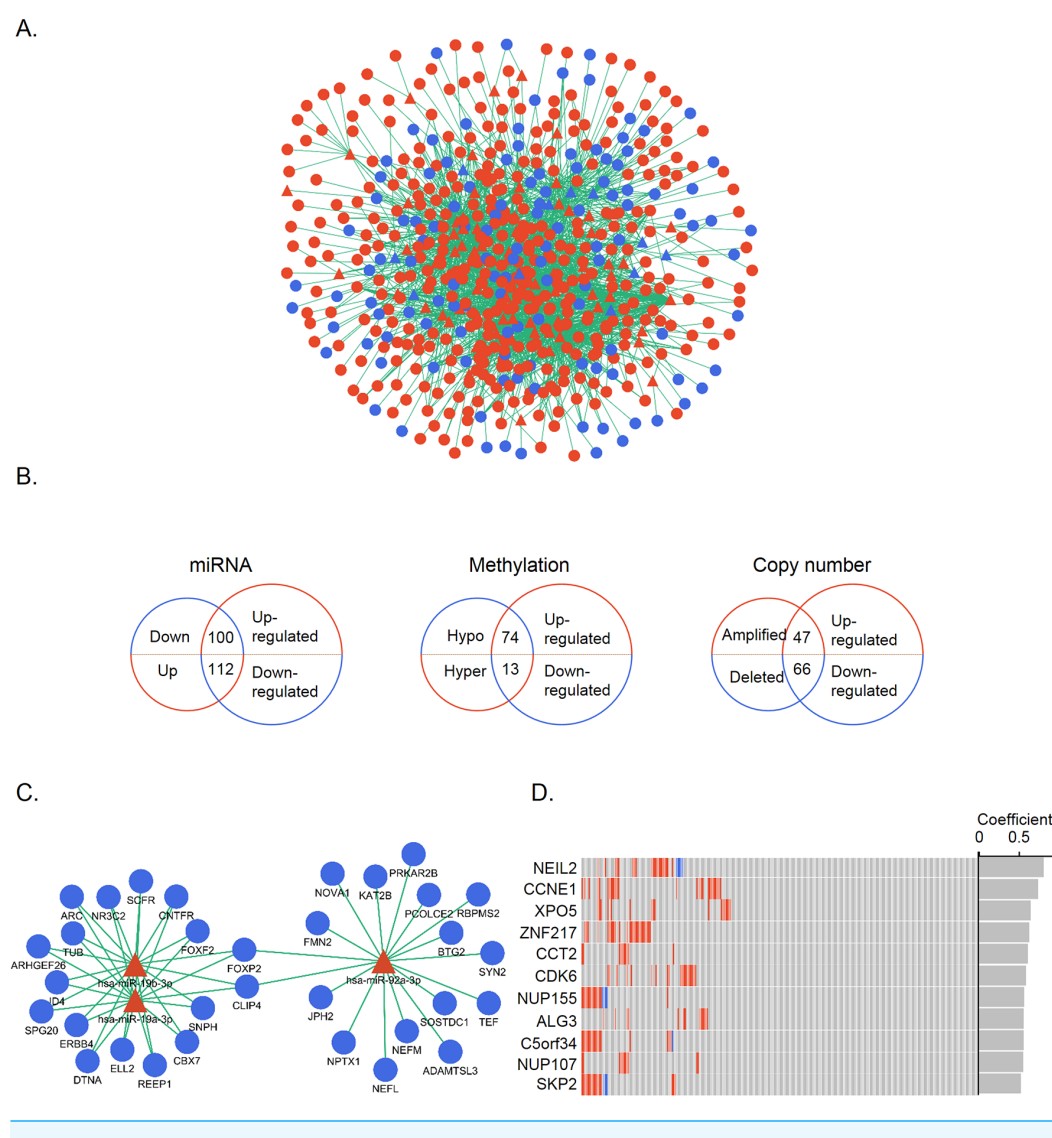

**Figure 3** **Identification of candidate genes controlled by multiple mechanisms.** (A) MiRNA-target gene network. Triangles and circles represent differentially expressed miRNAs and genes, respectively; red and blue represent up- and down-regulation, respectively. (B) Venn diagrams illustrating the three groups of candidate genes controlled by different regulatory mechanisms. Small and large circles represent altered regulatory controls and genes, respectively. (C) MiR-19a-3p, miR-19b-3p, and miR-92a-3p and their inversely correlated targets. Red and blue represent up- and down-regulation, respectively. (D) CNA distribution of significant genes ($r > 0.5$) in 272 STAD patients. Amplifications and homozygous deletions are labeled with red and blue, respectively. Copy number and gene expression correlation coefficients are shown in the gray bar.

$r > 0.05$ and $P < 0.05$ (Fig. 3D). NEIL2 and CCNE1 were identified as having the highest correlation ($>0.7$), indicating that the expression of these two genes was strongly influenced by CNA. NEIL2 is located at 8p23.1, a region that has been linked to tumorigenesis and patient prognosis (*Goh et al., 2011*; *Frankel et al., 2014*); CCNE1 is located at 19q12, and its amplification has been shown to promote tumor cell proliferation (*Leung et al., 2006*).

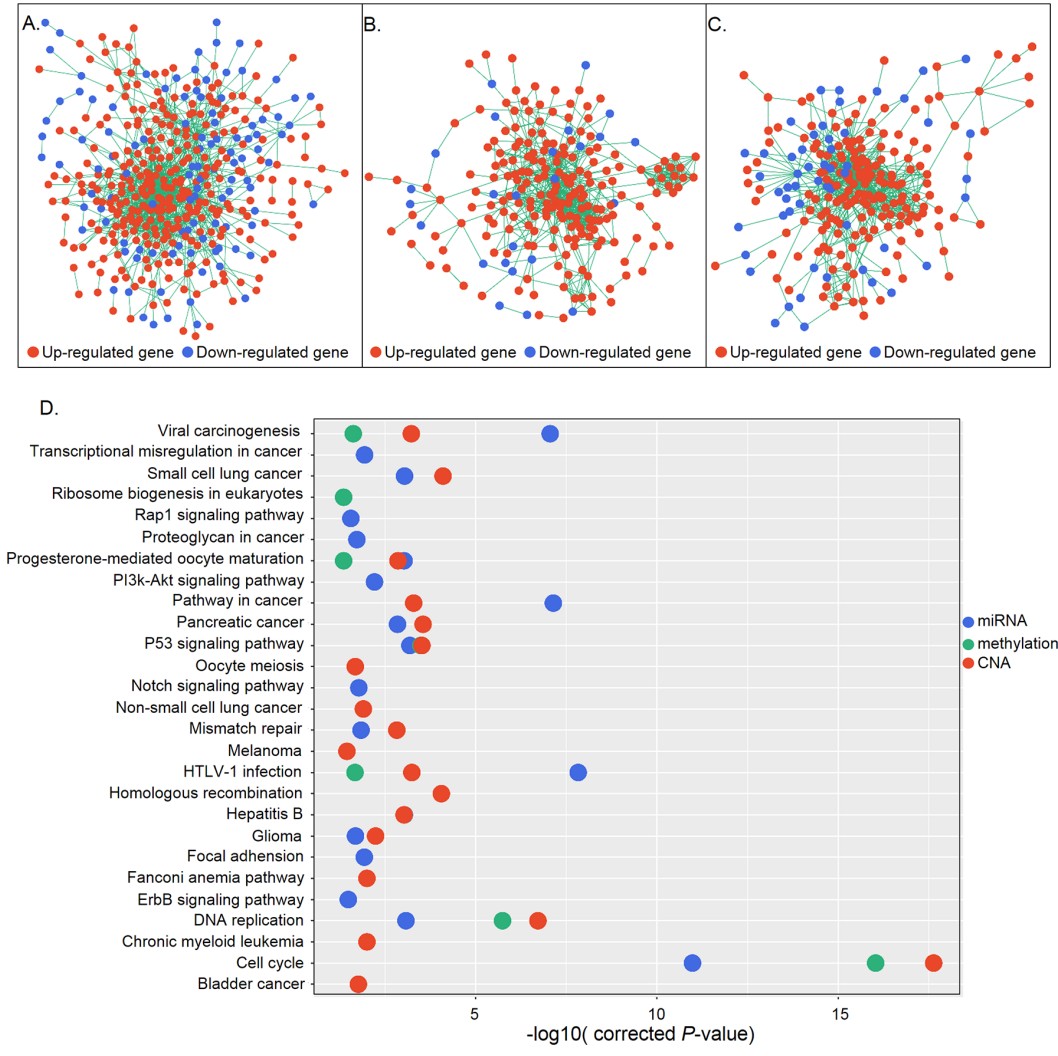

**Figure 4** **Gene interaction networks according to regulatory mechanisms.** (A–C) Gene interaction networks under control of miRNAs (A), methylation (B), and CNA (C). Red and blue represent up- and down-regulation, respectively. (D) KEGG pathway analysis of genes in the three networks. The *X* and *Y* axes show the −log10 (corrected *P* value) and name of each pathway, respectively. MiRNA-, methylation-, and CNA-regulated groups are labeled in blue, green, and red, respectively.

## Gene network construction according to regulatory mechanisms

DEGs are not independent from each other but form regulatory networks under the control of different factors. Also, only a small group of DEGs are directly regulated by regulatory changes; most are indirectly controlled, in some cases regulated by those directly affected genes (*Mine et al., 2013*). Based on this information, we generated networks under different regulatory mechanisms consisting of candidate genes that were directly regulated and their differentially expressed first neighbors. A total of 415, 228, and 233 genes constituted three networks which were under control of miRNA, methylation, and CNA, respectively (Figs. 4A–4C). Functional enrichment analysis revealed that genes in the miRNA regulatory network were mainly associated with the cell cycle, P53 signaling, DNA

**Table 2** Information on nine key genes.

| Gene | Expression | miRNA[a] | Methylation | CNA | Pathway |
|------|-----------|----------|-------------|-----|---------|
| SRC | Up | 1 | | | Focal adhesion |
| KAT2B | Down | 9 | | | Notch signaling |
| NR3C1 | Down | 11 | | | |
| CDK6 | UP | 2 | | Amplified | Cell cycle |
| MCM2 | Up | | Hypo | | Cell cycle |
| PRKDC | Up | | Hypo | | Cell cycle |
| BLM | Up | | | Amplified | Homologous recombination |
| CCNE1 | Up | | | Amplified | Cell cycle |
| PARK2 | Down | | | Deleted | |

**Notes.**
[a] Number of regulatory miRNAs.
CNA, copy number alteration; Down, down-regulated; Hypo, hypomethylation; Up, up-regulated.

replication, HTLV-1 infection, and cancer-related signaling pathways, whereas those in the methylation regulatory network were mainly involved in the cell cycle, P53 signaling, and DNA replication. Genes in the CNA regulatory network were also associated with the cell cycle- and cancer-related signaling pathways (Fig. 4D). Cell cycle-related pathways (i.e., cell cycle, P53 signaling, and DNA replication), viral carcinogenesis, HTLV-1 infection, and the progesterone-mediated oocyte maturation pathway were common to the three groups. The identification of cell cycle associated pathways indicates that aberrant cell cycle control is a critical feature of gastric cancer, as it is for most tumors (*Sherr, 1996*).

## Topological analysis and key gene identification

To identify hub genes in each network, we calculated the parameters of degree, betweenness, and closeness for each node (Table S4). For the miRNA regulatory network, the candidate genes SRC, KAT2B, and NR3C1 were identified as hubs, as they ranked top 10 under at least two parameters. In the methylation regulatory network, candidate genes MCM2 and PRKDC were identified as hubs. In the CNA regulatory network, the candidate genes BLM, CCNE1, and PARK2 were hubs. These eight genes were regulated by upstream regulatory controls and in turn acted on downstream effectors, and could therefore be taken as key genes (Table 2). Furthermore, as the candidate gene CDK6 ranked top 10 in two networks (miRNA and CNA controlled networks) for its degree, we also took it as key gene. CDK6 encodes a cell cycle regulatory protein and is frequently amplified in gastric cancer (*Ooi et al., 2017*). We found here that CDK6 was regulated by miR-137 and miR-145-5p, consistent with previous reports (*Shao et al., 2013*; *Zheng et al., 2015*). Interestingly, four of the identified key genes (CDK6, MCM2, PRKDC, and CCNE1) are on the cell cycle pathway, underscoring the importance of this process in gastric cancer (Table 2).

## Genes in the disease-associated network

To assess the biological significance of the identified genes in gastric cancer, we searched the literature and constructed a gastric cancer associated network using the nine key genes as input (Fig. 5). There were 76 nodes with 235 interactions in the network; nearly one-third of the nodes were DEGs identified in this study, while seven of the nine identified key genes
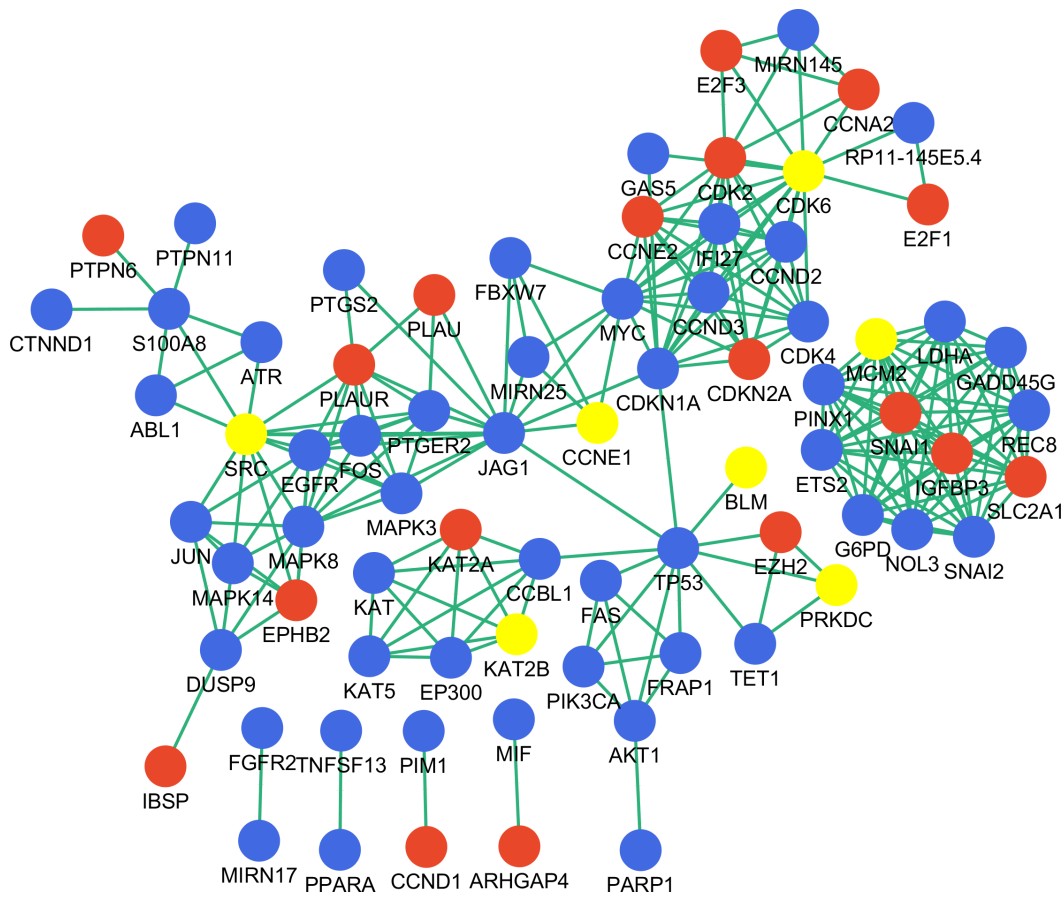

**Figure 5** **Disease-associated gene network constructed with key genes.** Red and blue represent DEGs and non-DEGs, respectively. Key genes are highlighted.

were highlighted in the network as being important in gastric cancer. However, NR3C1 and PARK2 were not included in the network, and therefore their function in gastric cancer requires further investigation.

## DISCUSSION

Gastric cancer is one of the most common and deadly malignancies; understanding the underlying molecular mechanisms is critical for developing more effective treatments. In this study, we used gene expression, miRNA expression, methylation, and copy number data to clarify the molecular etiology of gastric cancer, and identified critical pathways as well as nine key genes that are potential biomarkers and therapeutic targets for gastric cancer.

Altered regulation of gene expression programs leads to the expression of different cancer hallmarks (*Tan & Yeoh, 2015*). However, studying DEGs does not provide the mechanism basis for their dysregulation. In contrast, an integrated analysis that combines transcriptome and other data can help to identify regulatory cascades and provide a more detailed understanding of cancer etiology (*Chari et al., 2010*). Several studies have

examined carcinogenesis by combining genomic, epigenetic, transcriptomic, and post-transcriptomic data (*Sun et al., 2011*; *Setty et al., 2012*; *Wang et al., 2014*). Here, we used a bimodal strategy to evaluate the interaction of gene expression alteration with regulatory changes (miRNA expression, DNA methylation, and copy number). Three groups of candidate genes were identified whose expression were altered along with that of regulatory controls. Our correlation analysis also revealed up-regulation of miR-92a-3p, miR-19a-3p, and miR-19b-3p and amplification of NEIL2 and CCNE1; these are known to be associated with carcinogenesis and thus validated our analytical approach.

While detecting alterations in gene expression resulting from the perturbation of multiple regulatory mechanisms is useful for comprehensive refinement of DEGs, the most challenging task is identifying genes that cause key changes. However, traditional experimentation to identify significant pathways and genes and their cause-effect relationships is labor-intensive. In contrast, a network-based approach provides a data-reduction scheme that limits the analysis to several related genes. Interaction networks are useful for studying the molecular pathology of diseases, with hub genes serving as key regulators (*Lefebvre et al., 2010*). Moreover, highly reliable results can be obtained by incorporating regulatory mechanisms into the network (*Dutta et al., 2012*). In the present study, interaction networks were generated based on differential gene expression and regulatory alterations. The functional enrichment analysis highlighted the importance of cycle-related pathways (i.e., cell cycle, P53 signaling, and DNA replication), viral carcinogenesis, HTLV-1 infection, and the progesterone-mediated oocyte maturation pathway in gastric cancer, while topological analysis identified nine candidate genes that modulated downstream genes and pathways and can therefore be considered as key regulators.

The identification of cell cycle pathway is not surprising, as its dysregulation leads to uncontrolled proliferation which is an important feature of cancer (*Sherr, 1996*). Activated P53 signaling pathway results in cell apoptosis or growth arrest, while dysregulation of P53 signaling has considerable impact on the process of carcinogenesis, as it increases the chances of tumor cell surviving progressively adverse conditions (*Evan & Vousden, 2001*). DNA replication is a highly regulated process that guarantees the faithful duplication of the genome during cell cycle. As genomic instability is an important hallmark of cancer, DNA replication is the most vulnerable cellular process that can lead to it (*Gaillard, García-Muse & Aguilera, 2015*). Viral carcinogenesis refers to cancer induced by a given virus, which can lead to malignant transformation of cells (*Volinia et al., 2006*). Recent studies on Epstein–Barr virus and gastric cancer have provided evidences to support this hypothesis on the progression of gastric cancer (*Camargo et al., 2016*; *Zhang et al., 2017*). HTLV-1 is a retrovirus which has been implicated in the occurrence of adult T-cell leukaemia and tropical spastic paraparesis. *Zhang et al. (2016b)* also identified HTLV-1 signaling pathway in gastric cancer, but the role of HTLV-1 in gastric cancer still needs further study. Progesterone-mediated oocyte maturation is a nongenomic signaling mediated by steroids. Nevertheless its role in gastric carcinogenesis remains unclear.

The identified key genes were divided into two groups that are either associated with cell cycle regulation or not. Among the former group, CDK6 and CCNE1 regulate the G1/S

phase of cell cycle; MCM2 is involved in DNA synthesis; and PRKDC plays a critical role in the DNA damage response and maintenance of genomic stability. All of these genes have been previously implicated in gastric cancer as biomarker or therapeutic target (*Liu et al., 2013*; *Li et al., 2013*; *Zheng et al., 2015*; *Zhang et al., 2016a*).

Among genes not involved in cell cycle pathway, SRC is a known oncogene that links signaling pathways controlling cell proliferation, invasion and angiogenesis. Inhibitors of SRC kinase activity have been used to treat gastric cancer (*Nam et al., 2013*). KAT2B is a histone lysine acetyltransferase that regulates gene transcription via interaction with p300/CREB-binding protein (*Zhu et al., 2009*); down-regulation of KAT2B promotes intestinal-type gastric cancer progression and is correlated with a poor clinical outcome (*Ying et al., 2010*). NR3C1 encodes glucocorticoid receptor which is associated with several biological process including inflammation and differentiation. While NR3C1 is reported to be down-regulated in primary gastric cancer, its function in gastric cancer needs more exploration (*Chang et al., 2009*). BLM is a RecQ family helicase that plays a critical role in homologous recombination repair (*Chu & Hickson, 2009*). PARK2 is frequently deleted in human tumors and is a tumor suppressor gene, although the precise role of PARK2 in gastric cancer requires more detailed investigation (*Gong et al., 2014*; *Hu et al., 2016*).

There were some limitations in this study. Firstly, the incomplete paired data for tumor and normal tissues may affect the reliability of the final results. Secondly, our study was based on omics data analysis and PPI network analysis; therefore, biological experiments are required to validate the findings.

In conclusion, this study identified key genes and pathways for gastric cancer through a network-based approach that combined multi-dimensional data. Despite some limitations, our findings nonetheless provide a set of potential biomarkers and drug targets for gastric cancer.

## ACKNOWLEDGEMENTS

We would like to acknowledge the work of the TCGA Research Network (http://cancergenome.nih.gov/), who generated the data used here.

### Funding

This work was supported by National Natural Science Foundation of China grants (No.81372295 and No.81402374), and a China National High Technology Research and Development Program grant (#2014AA020604). The funders had no role in study design, data collection and analysis, decision to publish, or preparation of the manuscript.

### Grant Disclosures

The following grant information was disclosed by the authors:
National Natural Science Foundation of China: 81372295, 81402374.
China National High Technology Research and Development Program: #2014AA020604.

## Competing Interests

The authors declare there are no competing interests.

## Author Contributions

- Wu Ren performed the experiments, analyzed the data, wrote the paper.
- Wei Li performed the experiments, prepared figures and/or tables.
- Daguang Wang wrote the paper, prepared figures and/or tables.
- Shuofeng Hu analyzed the data, contributed reagents/materials/analysis tools.
- Jian Suo conceived and designed the experiments, reviewed drafts of the paper, collected fund.
- Xiaomin Ying contributed reagents/materials/analysis tools, reviewed drafts of the paper, collected fund.

## Data Availability

All data were re-analyzed from the publicly available dataset Broad GDAC Firehose http://gdac.broadinstitute.org.

## Supplemental Information

Supplemental information for this article can be found online at http://dx.doi.org/10.7717/peerj.3385#supplemental-information.

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
