# Peer review of "Combining multi-dimensional data to identify key genes and pathways in gastric cancer"

_PeerJ, doi:10.7717/peerj.3385_

## Round 0.1 · original submission · Minor Revisions

Dear Authors,

As you can see below the referees suggest a number of small changes that should be incorporated in the revised version.

Furthermore, they recommend to extend the analysis and discussion in an attempt to identify more clear links with gastric cancer beyond the currently identify genes and functions, that seems to correspond to general cancer related functions, such as proliferation and cell cycle.

·

Basic reporting

1)This paper is written in standard English, it is clear and I will only mention a few points where I would make changes:

- Throughout the text the authors speak of 'bimodal interactions' to refer to consistent combination of up/down regulated regulators and targets. I am not sure this term is appropriate. Maybe these could be called just regulatory interactions?
- Line 49, 3' untranslated region
- Line 68, change ; for ,
- Line 99 Genomic should be genomic
- Line 123 the sentence is unclear
- Line 187 I think there is a problem, if we eliminate the inversely correlated pairs how can we impose r<-0.1
- Line 194 unclear 'After remaining'
Line - 290 PPI network centralization - not sure what this means
- Figure 3 caption: labeled with red and blue (not on)

2) Literature is cited appropriately - at least on the clinical aspects of the work
3) The data is provided with the code and the structure is ok

Experimental design

The question addressed is whether integration of multiple datasets for gastic cancer can identify genes of interest and their regulators as possible biomarkers or drug targets for this disease.

- Line 145 the power law of the degree distribution is mentioned but not used. Only the degree is considered in the end. Also the network has some directed edges which are not taken into account in the calculation

Gene ontology analysis with DAVID should be updated to a newer version of the tool (6.7 is from 2010).

The analysis seems correctly performed, it is very basic. It would be nice if the authors could expand the network analysis a bit more. Just seeing genes that are connected is only one aspect of the network topology (could consider betweenness, difference between in-degree/out-degree, clustering coefficient).

Validity of the findings

- Line 184 MYEF2 no mentioning of importance of this finding, could be interesting
- Line 215 no mentioning of significance of findig progesterone mediated oocyte maturation
Unfortunately nothing major can be concluded by this analysis other than the fact that the cell cycle and srk are important in gastric cancer. The authors could compare with finding in other gastric cancer papers or delve a bit deeper on the few findings that are not facts already known in the literature (cancer is associated to cell cycle and DNA damage...).

There is no discussion about whether the observation of interactions on the 3 levels matches what one would expect. Whereas it is clear that methylation at promoters can in general be associated to reduced gene expression, it is not clear at all whether miRNA and CNA regulation should be in one direction or the other. miRNAs appear to be regulating directly or inveresely a similar number of genes (100 vs 112) and the same is true for CNA to a lesser extent (47 vs 66). The authors could comment on this more.

Additional comments

The work is correct and reproducible based on the material provided.
The authors should make the suggested modifications to the text to improve its clarity.
However, the authors could make more effort to further interpret their findings or provide a more extensive analysis. The claim that they find biomarkers for gastric cancers is in part justified (they do identify genes that could be biomarkers) but they are likely to be genes that are markers for any proliferative state (cell cycle). More effort could be made to dig in these networks for some novel candidate biomarkers or to dissect unknown aspects of their regulation.

·

Basic reporting

a. In line 43, it is better to say “improving gastric cancer outcome”.
b. In line 271-272, it is clearer to say “the identified regulated genes were divided into 2 groups and each group contained 4 genes that are either associated with cell cycle regulation or not”.
c. In line 278, it is better to say “genes not involved in cell cycle pathway”.
d. In the line 289, it is better to change “may have affected” to “may affect”.

Experimental design

a. It’s better to have a more specific aspect rather than a simply list of all the information.
b. The methods are described with sufficient detail and information.

Validity of the findings

Data and statistical analysis are well demonstrated.

Additional comments

Authors have provided a significant work of screening candidate key genes and pathways potentially involved in gastric cancer, although it requires a huge amount of work in future to identify their functions. For one thing, I suppose that it will be more interesting if authors could dig into genes concerning one specific aspect of gastric cancer to get more solid and confirmed data in future study. For another, authors need to pay more attention to the language to convey clearer and more accurate ideas to readers.

---

## Round 0.2 · accepted · Accept

Dear Authors,

Thanks for addressing the reviewers comments and suggestions. The addition of the new network analysis data, new version of David, together with other small changes and additions makes the paper clearer and easier to follow.